# Soil Collected from a Single Great Smoky Mountains Trail Contains a Diversity of *Listeria monocytogenes* and *Listeria* spp.

Michelle L. Claxton,[a] Lauren K. Hudson,[a] Daniel W. Bryan,[a] Thomas G. Denes[a]

[a]Department of Food Science, University of Tennessee, Knoxville, Tennessee, USA

Michelle L. Claxton and Lauren K. Hudson contributed equally to this work. Author order was determined based on alphabetical order.

**ABSTRACT** *Listeria monocytogenes*, a foodborne pathogen, and other *Listeria* spp. are present in natural environments. Isolating and characterizing strains from natural reservoirs can provide insight into the prevalence and diversity of *Listeria* spp. in these environments, elucidate their contribution to contamination of agricultural and food processing environments and food products, and lead to the discovery of novel species. In this study, we evaluated the diversity of *Listeria* spp. isolated from soil in a small region of the Great Smoky Mountains National Park, the most biodiverse national park in the U.S. National Park system. Of the 17 *Listeria* isolates recovered, whole-genome sequencing revealed that 14 were distinct strains. The strains represented a diversity of *Listeria* species (*L. monocytogenes* [$n = 9$], *L. cossartiae* subsp. *cossartiae* [$n = 1$], *L. marthii* [$n = 1$], *L. booriae* [$n = 1$], and a potentially novel *Listeria* sp. [$n = 2$]), as well as a diversity of sequence types based on multilocus sequence typing (MLST) and core genome MLST, including many novel designations. The isolates were not closely related ($\geq 99.99\%$ average nucleotide identity) to any isolates in public databases (NCBI, PATRIC), which also indicated novelty. The *Listeria* samples isolated in this study were collected from high-elevation sites near a creek that ultimately leads to the Mississippi River; thus, *Listeria* present in this natural environment could potentially travel downstream to a large region that includes portions of nine southeastern and midwestern U.S. states. This study provides insight into the diversity of *Listeria* spp. in the Great Smoky Mountains and indicates that this environment is a reservoir of novel *Listeria* spp.

**IMPORTANCE** *Listeria monocytogenes* is a foodborne pathogen that can cause serious systemic illness that, although rare, usually results in hospitalization and has a relatively high mortality rate compared to other foodborne pathogens. Identification of novel and diverse *Listeria* spp. can provide insights into the genomic evolution, ecology, and evolution and variance of pathogenicity of this genus, especially in natural environments. Comparing *L. monocytogenes* and *Listeria* spp. isolates from natural environments, such as those recovered in this study, to contamination and/or outbreak strains may provide more information about the original natural sources of these strains and the pathways and mechanisms that lead to contamination of food products and agricultural or food processing environments.

**KEYWORDS** Great Smoky Mountains National Park, *Listeria*, *Listeria monocytogenes*, *Listeria* spp.

Address correspondence to Thomas G. Denes, tdenes@utk.edu.

The authors declare no conflict of interest.

Members of the genus *Listeria* are characterized as Gram-positive, short rods that are motile via peritrichous flagella and have aerobic or facultative anaerobic metabolism (1). The *Listeria* genus currently contains 26 validly published species (as of October 2021) (2, 3). Of these, only two are typically described as pathogenic to humans: *Listeria monocytogenes*, a human and animal pathogen, and *Listeria ivanovii*,

which primarily infects ruminants but is an extremely rare opportunistic human pathogen (1, 4–9). Two other species, *Listeria seeligeri* and *Listeria innocua*, have under rare circumstances been documented as the causative agent of human illness (10, 11). Listeriosis in humans is primarily caused by ingesting food contaminated with *L. monocytogenes* (1). Compared to other foodborne pathogens, the incidence of listeriosis is low (0.3 cases per 100,000 population); however, the proportions of cases resulting in hospitalization (98%) or death (16%) are high (12), as are the economic costs associated with this pathogen (estimated as up to $8.6 billion) (13).

As *L. monocytogenes* is a foodborne pathogen, much research has focused on evaluating the distribution of *L. monocytogenes* and other *Listeria* spp. in food, food processing facilities, agricultural settings, and animals. However, *Listeria* spp. are distributed in a variety of natural environments, including soil, sewage, animal feces, agricultural environments, decaying vegetation, and water (1, 14–17). The adaptability of this microorganism plays a role in its wide distribution: *Listeria* spp. can survive in a wide range of pHs (4.5 to 9.2), temperatures (0 to 45°C), and salt concentrations ($\leq$10% NaCl) (18). The extensive variety of natural reservoirs causes variable genomic flexibility and recombination (19). There is interest in describing the distribution, prevalence, variability, diversity, and characteristics of *Listeria* spp. from natural environments (17). A better understanding of the ecology and evolution of *Listeria* spp. will help to elucidate the sources and pathways that lead to contamination of agricultural products or food processing facilities (20). Additionally, sampling in agricultural and natural environments can lead to the discovery of novel *Listeria* spp. Since 2010, 12 novel *Listeria* spp. isolated from natural environments have been described (14, 21–24), and this indicates that novel species of *Listeria* strains can be found by surveying biodiverse natural reservoirs.

A recent study examined *Listeria* spp. in soil across the contiguous United States and reported a 31% prevalence of *Listeria* spp. in the soil samples collected, with the highest prevalence throughout and near the Mississippi River Basin (15). Additionally, those researchers found that isolates fell into 12 genetic phylogroups (consisting of nine different species, with *L. monocytogenes* separated into three phylogroups and *Listeria seeligeri* into two). The most prevalent *Listeria* phylogroup was L8 (*Listeria welshimeri*), followed by L1 (*L. monocytogenes* lineage III) and L12 (*Listeria booriae*) (*L. monocytogenes* would be the most prevalent if all three lineage phylogroups had been considered as a whole) (15). Lao et al. found that genomic flexibility (including pan-genome openness, the extent of positive selection, and the frequency of recombination) correlated with phylogroup prevalence and indirectly with habitat breadth. The geographical distance between strains has also been shown to correlate with gene content difference in other genera, but only minimally in *Listeria* spp. (15, 19). In a study conducted in Australia, *L. seeligeri*, *L. innocua*, and *L. ivanovii* were the most dominant species isolated from soil and water samples (25).

One ideal sampling location for isolating and evaluating the diversity of *Listeria* spp. is the Great Smoky Mountains, a subrange of the Appalachian Mountains that is located along the Tennessee-North Carolina border and mostly contained within Great Smoky Mountains National Park (GSMNP). The GSMNP contains a high level of biodiversity: it is the most biodiverse of all the parks in the U.S. National Park system and contains more than 19,000 documented species of animals, plants, fungi, and other organisms within the over 800-square mile (2,071.99-km²) area, with an estimated additional 80,000 to 100,000 undocumented species (26).

The purpose of this study was to evaluate the diversity of *Listeria* spp. isolated from soil samples in a small region of the GSMNP adjacent to Twentymile Creek. The downstream flow of this creek includes prominent water systems, including the Tennessee and Mississippi Rivers, and runs through nine southeastern and midwestern states (27, 28). *Listeria* spp. present in this location may potentially be carried downstream to large areas of the region that contain people, livestock, farms, and food processing facilities.

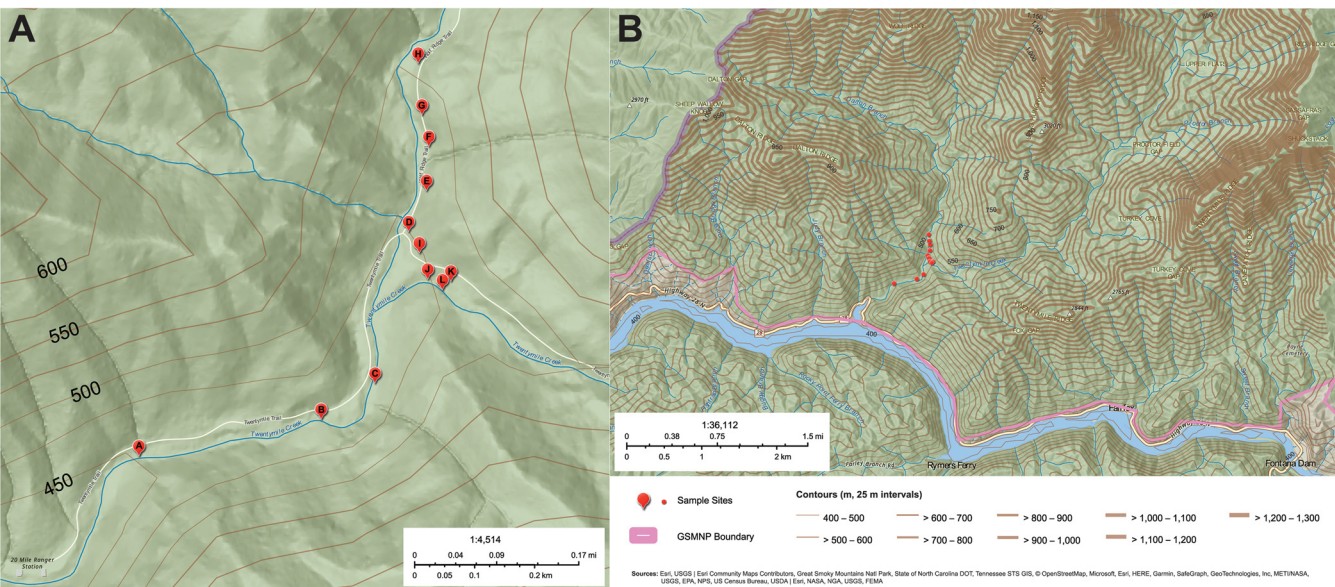

**FIG 1** Maps of sampling location and features of the surrounding area. (A) Twentymile Loop trail with sampling sites (letters A to L) indicated with red circles. (B) Surrounding area around the sampling location. Topographical contours with an interval of 25 m are shown as brown lines, with higher elevations indicated by thicker lines. The maps were created with ArcGIS.

## RESULTS AND DISCUSSION

Twelve soil samples were collected from 12 separate sites along the Twentymile Trail in GSMNP (Fig. 1). A total of 17 *Listeria* spp. isolates were recovered from nine of the samples (Table 1). The isolates were sequenced, and the resulting assemblies ranged from 2.81 to 3.78 Mb in length and 37.64 to 41.78% G+C content, which are consistent for *Listeria* spp. (29, 30). All genome assemblies were estimated as ≥96.69% complete and ≤1.10% contamination (see Table S1 in the supplemental material), indicating near completeness and low to no detectable contamination (31). All but three isolates had average nucleotide identity (ANI) values of >96% with their respective type strains (Fig. 2), which was above

**TABLE 1** *Listeria* spp. isolates recovered from soil samples collected in Great Smoky Mountains National Park

| Isolate | Species | Site | Phylo-genetic lineage | Sublineage | PCR sero-group | MLST | CC | cgMLST type |
|---|---|---|---|---|---|---|---|---|
| UTK C1-0001 | *L. booriae* | A | Other[d] | NA[e] | L | 2852[g] | ST2852 | NA |
| UTK C1-0002[a] | *L. cossartiae* subsp. *cossartiae* | B | Other | NA | L | 2853[g] | ST2853 | NA |
| UTK C1-0003 | *L. monocytogenes* | C | II | SL2496 | IIa | 2496 | ST2496 | CT10967[g] |
| UTK C1-0004 | *L. monocytogenes* | A | II | SL1875 | AT[f] | 1875 | ST1875 | CT10974[g] |
| UTK C1-0005[a] | *L. cossartiae* subsp. *cossartiae* | B | Other | NA | L | 2853[g] | ST2853 | NA |
| UTK C1-0006 | *L. monocytogenes* | C | II | SL2854 | IIa | 2895[g] | ST2894 | CT10970[g] |
| UTK C1-0007[b] | *L. monocytogenes* | F | II | SL2295 | AT | 2295 | ST2295 | CT10966[g] |
| UTK C1-0012[b] | *L. monocytogenes* | F | II | SL2295 | AT | 2295 | ST2295 | CT10966[g] |
| UTK C1-0013 | *L. monocytogenes* | L | II | SL2855 | AT | 2855[g] | ST2855 | CT10969[g] |
| UTK C1-0014 | *L. marthii* | H | Other | NA | L | 2856[g] | ST2856 | NA |
| UTK C1-0015 | *Listeria* spp. | I | Other | NA | L | 2857[g] | ST2857 | NA |
| UTK C1-0016 | *L. monocytogenes* | H | II | SL2858 | AT | 2858[g] | ST2858 | CT10973[g] |
| UTK C1-0017[c] | *Listeria* spp. | J | Other | NA | L | 2859[g] | ST2859 | NA |
| UTK C1-0018 | *L. monocytogenes* | K | II | SL2860 | AT | 2860[g] | ST2860 | CT10968[g] |
| UTK C1-0021 | *L. monocytogenes* | L | II | SL2861 | AT | 2861[g] | ST2861 | CT10972[g] |
| UTK C1-0023 | *L. monocytogenes* | K | II | SL2862 | L | 2862[g] | ST2862 | CT10971[g] |
| UTK C1-0024[c] | *Listeria* spp. | J | Other | NA | L | 2859[g] | ST2859 | NA |

[a]Isolates were identical (>99.99% ANI and hqSNP distance of 0).
[b]Isolates were identical (>99.99% ANI and hqSNP distance of 0).
[c]Isolates were identical (>99.99% ANI and hqSNP distance of 0).
[d]Other *Listeria* species.
[e]NA, not applicable.
[f]Atypical.
[g]Novel designation.

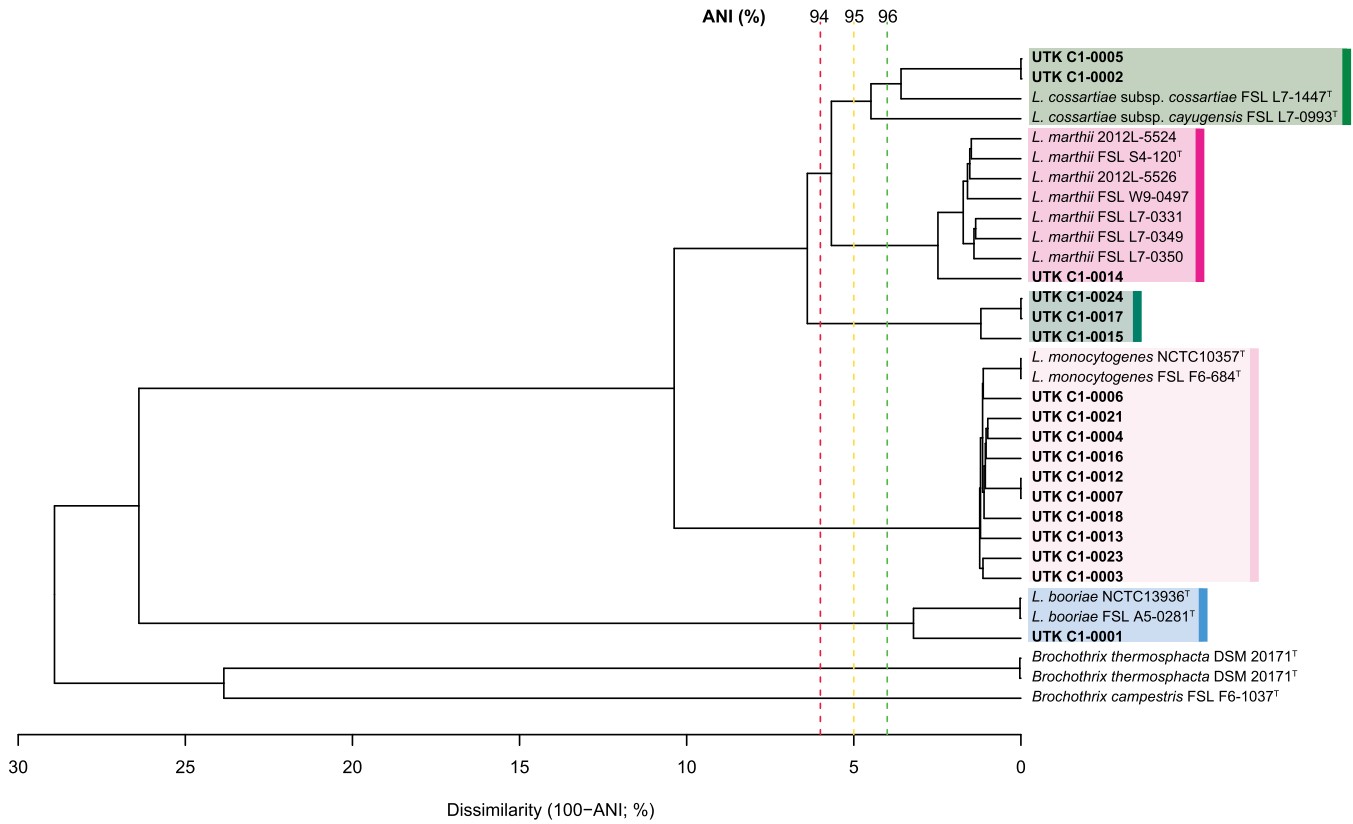

**FIG 2** Average nucleotide identity (ANI) dendrogram of study isolates and type strains. The dendrogram includes isolates from this study (bold), *Listeria* spp. type strains (indicated with a superscript letter T after the strain name), and three *Brochothrix* spp. type strains (included as an outgroup). Dashed vertical lines indicate dissimilarities of 6% (red), 5% (yellow), and 4% (green), which corresponded to ANI values of 94%, 95%, and 96%, respectively. Species clusters are indicated by boxes.

the 95% cutoff typically used for bacterial species distinction (32, 33). Of the 17 isolates, 14 were distinct strains (those that were >99.99% ANI and with 0 high-quality single-nucleotide polymorphisms [hqSNPs] apart were considered the same strain). The most prevalent species was *L. monocytogenes* ($n$ = 10), with nine distinct strains (UTK C1-0007 and UTK C1-0012 were identical). Other known species that were isolated included *L. cossartiae* subsp. *cossartiae* ($n$ = 2; one distinct strain; UTK C1-0002 and UTK C1-0005 were identical), *L. marthii* ($n$ = 1), and *L. booriae* ($n$ = 1). Three of the isolates (two distinct strains; UTK C1-0017 and UTK C1-0024 were identical) clustered together but shared <95% ANI with all known *Listeria* spp. type strains, suggesting they would qualify as a novel species. After the initial submission of this work, these novel species isolates were further characterized and determined to belong to the proposed novel species "*Listeria swaminathanii*" (34, 35).

The genomic assemblies were further *in silico* characterized and subtyped. Of the *L. monocytogenes* strains, all were lineage II and the PCR serogroup designations (36–38) consisted of two IIa (which includes serotypes 1/2a and 3a), one L (which includes less common *L. monocytogenes* serotypes 4a, 4ab, and 4c and most other *Listeria* spp.), and six "atypical." Genetic lineage II contains serotypes 1/2a, 1/2c, and 3a (39). The other species were all PCR serogroup L. Each of the 14 distinct strains was assigned a different multilocus sequence typing (MLST) sequence type (ST), of which 11 were novel (Table 1). Similarly, each of the nine *L. monocytogenes* strains was assigned a different core genome multilocus sequence typing (cgMLST) ST, all of them novel. None of the *L. monocytogenes* isolates clustered with any other isolates in the NCBI Pathogen Detection Isolates browser. Preliminary searches (BLAST, PATRIC) for closely related isolates yielded 36 isolates from a variety of sources for further evaluation using ANI. UTK C1-0003 had an ANI of 99.92 to 99.98% with strain FSL L7-0001 (Fig. 3), which was also isolated from soil in North Carolina (15). All other *L. monocytogenes* isolates had ANI

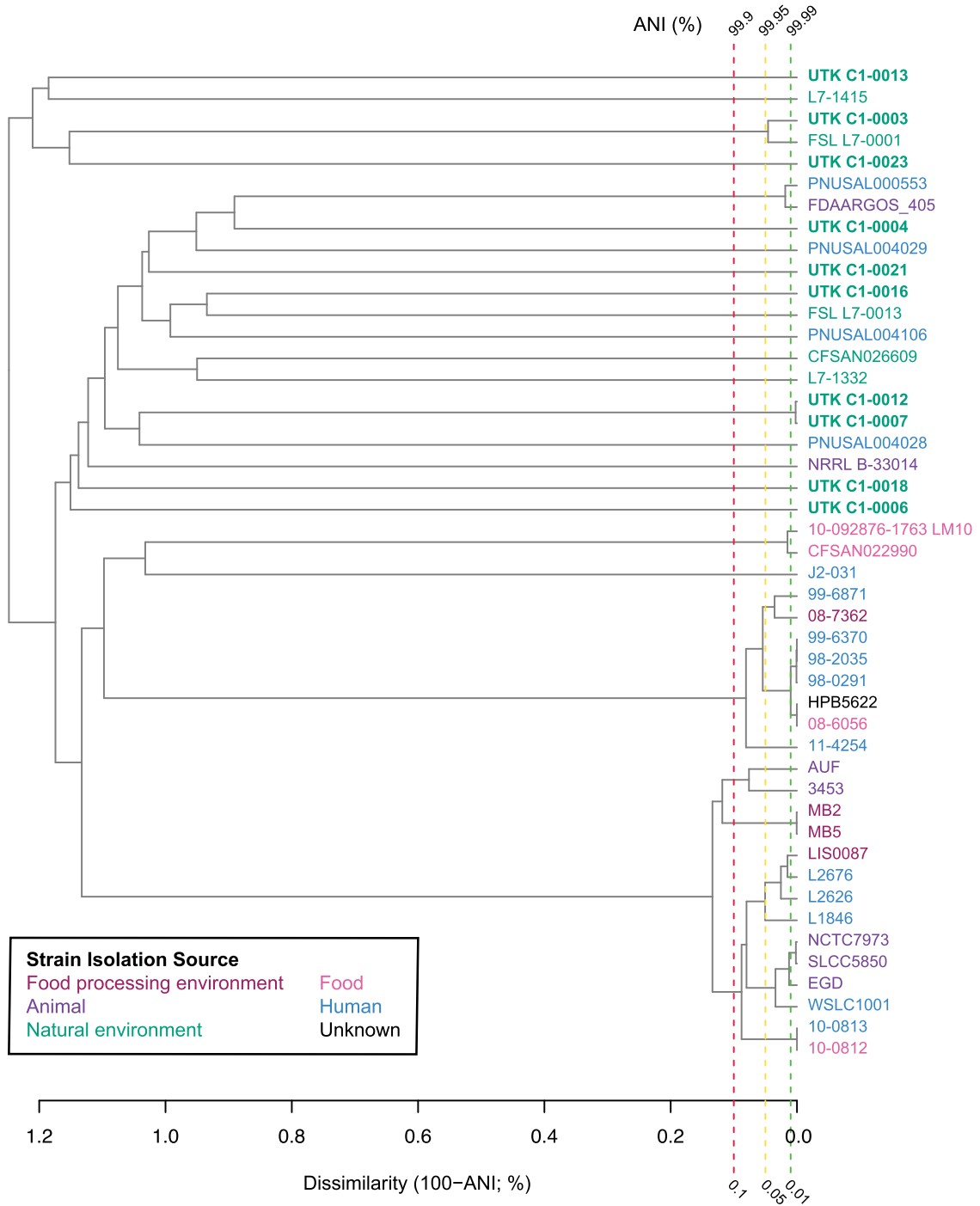

**FIG 3** Average nucleotide identity (ANI) dendrogram of *L. monocytogenes* study isolates and their most similar strains. The dendrogram includes *L. monocytogenes* isolates from this study (bold) and their most similar strains. Label color indicates strain isolation source: maroon, food processing environment; pink, food; purple, animal; blue, human; teal, natural environment; black, unknown. Dashed vertical lines indicate dissimilarities of 0.1% (red), 0.05% (yellow), and 0.01% (green), which corresponded to ANI values of 99.9%, 99.95%, and 99.99%, respectively.

values of ≤99.17% with their most similar strains. The lack of closely related isolates in public databases, along with the varied and novel ST assignments, indicated that the isolated strains represented a wide level of diversity and novelty within this genus.

Additionally, the assemblies were queried for genes associated with virulence. In total, 126 different virulence-associated genes were identified among the 17 isolates (Data Set S1). Only 11 of these were identified in all isolates, and 62 were identified in all isolates

except the *L. booriae* isolate. The 10 *L. monocytogenes* isolates each contained 106 to 110 virulence-associated genes, "*L. swaminathanii*" contained 81 to 89, *L. marthii* contained 81, *L. cossartiae* contained 80, and *L. booriae* contained 12. All *L. monocytogenes* isolates contained *inlA* and *inlB*, which encode internalins that promote entry into host cells (40, 41), and the other internalin genes *inlC*, *inlE*, *inlL*, *inlP*, *inlP3*, and *inlPq*; these were not identified in any of the other species. All *L. monocytogenes* isolates contained the six genes that comprise *Listeria* Pathogenicity Island 1 (LIPI-1; *actA*, *hly*, *mpl*, *plcA*, *plcB*, and *prfA*), with the exception of *actA* not being detected in UTK C1-0021. LIPI-1 gene products are essential for actin-based motility and the intracellular life cycle in pathogenic *Listeria* spp. (41–43). Seven of the eight LIPI-3 genes (*llsAGXBYDP*) were found in only two of the three "*L. swaminathanii*" isolates; the products of these genes are involved in the production of the hemolytic toxin listeriolysin S (LLS) (44). Typically, LIPI-3 is found in *L. monocytogenes* lineage I (41, 44, 45). All LIPI-4 genes (*LM9005581_70009* to *LM9005581_70014*) were only identified in both of the *L. cossartiae* isolates and in the *L. marthii* isolate. LIPI-4 has been associated with the hypervirulent *L. monocytogenes* CC4 clonal group and sublineage 4 (SL4) and closely related sublineages; it is involved in neural and placental infection (41, 45, 46).

The most prevalent species, *L. monocytogenes*, was recovered from six different sites throughout the sampling area (A, C, F, H, K, and L) (Table 1, Fig. 1), including sites at opposite sides of the sampling area (A and H, approximately 0.8 km apart). In contrast, the other species were each only recovered from a single site (*L. booriae*, *L. cossartiae*, and *L. marthii*) or from two neighboring sites (potential novel species). The species identified in the present study correspond to 4 of the 12 phylogroups from soil samples described by Liao et al. (15): L2 (*L. monocytogenes* lineage II), L4 (*L. marthii*), L5 (*L. cossartiae*), and L12 (*L. booriae*). In that previous study, phylogroups L2 and L12 had wide distribution areas that both contained GSMNP, with L12 being isolated from the GSMNP region specifically. In contrast, phylogroups L4 and L5 had much smaller distributions, with L4 only being isolated in Kentucky, Pennsylvania, and New York, and L5 only in Alabama (neither was isolated in the GSMNP region). In the Liao et al. study, the distribution areas of phylogroups L1 (*L. monocytogenes* lineage III), L3 (*L. monocytogenes* lineage I), L6 (*L. innocua*), and L8 (*L. welshimeri*) contained the GSMNP region, and L3 and L6 were actually isolated from the GSMNP region (15); however, none of these phylogroups was isolated in the present study, which may indicate that they were not present in the samples. Alternatively, these phylogroups may have been present in the samples but not recovered; for example, they could have been suppressed or outcompeted by other strains during the enrichment step (47–53), or their colonies may have simply not been selected off the modified Oxford agar plates for isolation and characterization.

*Listeria* spp. and other pathogens can be transported throughout the natural environment soil to water systems through flooding or rain events (25, 54–60). *Listeria* spp. and *L. monocytogenes* have been isolated from natural water sources (16, 61–64). All sample sites that tested positive for *L. monocytogenes* (sites A, C, F, H, K, and L) in the current study were within 50 ft (15.24 m) of Twentymile Creek. The downstream flow of this creek includes the Little Tennessee, Tennessee, Ohio, and Mississippi Rivers. Ultimately, the flow from Twentymile Creek to the Gulf of Mexico goes through nine states (North Carolina, Tennessee, Alabama, Kentucky, Illinois, Missouri, Arkansas, Mississippi, and Louisiana) (27, 28). This is significant, because *L. monocytogenes* from the sampling location could potentially enter the creek and be transported to local or regional agricultural environments downstream. This could lead to preharvest contamination of produce or agricultural environments via *L. monocytogenes* present in soil, wildlife carriers, raw or improperly composted manure, or irrigation water (61, 65, 66). None of the *L. monocytogenes* strains isolated in the current study were closely related to any isolates in public databases that were isolated from natural or agricultural environments downstream of the sampling site. This could have been due to a lack of total environmental sampling. More research is needed to better understand *L. monocytogenes* persistence in and transmission through the environment.

Furthermore, food processing environments can become contaminated with *Listeria* spp., which can lead to contamination of food products during processing. For environmental monitoring programs in food processing facilities, testing for *Listeria* spp. is typically conducted as an indicator for the potential presence of *L. monocytogenes* and/or for monitoring the effectiveness of sanitation practices (67–70). Once established in the food processing environment, *Listeria* spp. may be able to persist for years (67, 71). *Listeria* spp. can be initially introduced into a food processing facility via food products, ingredients, equipment, humans, etc. (72, 73), but often, the original source of the strain is unknown. Though none of the *L. monocytogenes* strains isolated in the current study were closely related to any isolates from clinical, food, or food processing environment sources in public databases, isolating and sequencing *L. monocytogenes* and *Listeria* spp. from natural environments, as was done in this and similar studies (15–17, 25), allows for comparison to contamination and/or outbreak strains. Doing so may provide more information about the original natural sources of these isolates and the pathways and mechanisms that lead to contamination of food products and agricultural or food processing environments.

This study further supported that natural environments, such as GSMNP, are reservoirs of *L. monocytogenes* and other *Listeria* spp. From just a small geographical area of the GSMNP, we were able to isolate a diversity of *Listeria*, which encompassed five species (four known and one potentially novel species), 14 distinct strains, and a variety of MLST and cgMLST ST designations. These strains were also unique, as evidenced by the novel ST designations and lack of closely related strains in public databases. Studies like this one contribute to the understanding of the diversity of this genus and can lead to the discovery of novel species. For future studies, a larger and broader geographical region should be investigated to determine ecological and evolutionary characteristics of *Listeria* spp. Furthermore, additional environmental factors could be assessed, such as soil moisture, soil pH, elevation, or season, to determine their relationships with *Listeria* spp. presence. Natural environments closer to agricultural locations should be investigated as well, to determine the potential for contamination events.

## MATERIALS AND METHODS

**Soil sample collection.** Soil samples were collected along the Twentymile Loop Trail in the Great Smoky Mountain National Park. Twelve soil specimens were collected from 12 different sites (Fig. 1A) along the trail. A Scientific Research and Collecting Permit (permit ID GRSM-2021-SCI-2152) issued by the U.S. Department of the Interior National Park Service to conduct field research in GSMNP was obtained prior to sample collection, and guidelines placed by the park were followed: collection from 15% or greater slopes and collection of less than 200 g of organic soil per site. Samples were collected with sterile scoops and placed into sterile bags (Whirl-Pak, Madison, WI) while wearing sanitized gloves. Sample bags were kept in soft coolers with ice packs until arrival at the lab (<4 h), where they were then stored at 4°C until processing. For each sample collection, site location coordinates, terrain description, time, and sample mass were documented.

***Listeria* spp. enrichment and isolation.** Methods used to isolate *Listeria* spp. from the soil samples were adapted from the FDA Bacteriological Analytical Manual (BAM) (74). For each sample, 25 g of soil was added to 225 mL of buffered *Listeria* enrichment broth (BLEB; BD Difco, Franklin Lakes, NJ) and stomached at 260 rpm for 1 min. The mixtures were then statically incubated at 30°C. After 4 h of incubation, 900 $\mu$L of *Listeria* selective enrichment supplement (LSES; Oxoid, Thermo Fisher Scientific, Waltham, MA) was added to each enrichment and incubation was continued at 30°C. At both 24 and 48 h of total incubation time, 1-mL aliquots of the BLEB enrichments were serially diluted to $10^{-3}$ in phosphate-buffered saline (PBS; 1 M; potassium chloride [Fischer Chemical], potassium phosphate [Acros Organics], sodium phosphate [Acros Organics], sodium chloride [Fischer Chemical]; pH 7.4 with 6 M HCl). A volume of 100 $\mu$L of each dilution was spread plated on modified Oxford agar (MOX; Oxford medium, Remel, Thermo Fisher Scientific, Waltham, MA, and modified Oxford antimicrobial supplement, BD Difco, Franklin Lakes, NJ) (24 total MOX plates) and incubated at 35°C for 24 h. A single colony characteristic of *Listeria* spp. (circular black colony surrounded by a black zone) was plucked from each MOX plate (23 total colonies) and added to 5 mL of brain heart infusion broth (BHI; BD Bacto, Franklin Lakes, NJ) and incubated at 25°C in a shaking water bath for 24 h. After incubation, freezer stocks were prepared by adding 1 mL of overnight culture to 1 mL of BHI broth with 30% (vol/vol) glycerol; these were stored at −80°C.

**DNA extraction, sequencing, and genome assembly.** DNA from the isolates (*n* = 23) was extracted using a Qiagen QIAamp DNA minikit (Hilden, Germany) per the manufacturer's protocol, with slight modifications, such as including an RNase treatment (75). DNA concentration and quality were measured using a NanoDrop spectrophotometer (Thermo Fisher Scientific, Waltham, MA) and Qubit system (Invitrogen, Thermo Fisher Scientific, Waltham, MA). Library preparation and sequencing were performed by the Microbial Genome Sequencing Center (Pittsburgh, PA). Sequencing libraries were prepared using Nextera XT kits (Illumina, San

Diego, CA). Sequencing was performed with the NextSeq 2000 platform, with 151-bp paired-end read chemistry. Raw reads were trimmed using Trimmomatic (v0.35, with the following parameters: Illuminaclip:NexteraPE-PE.fa, 2:30:10, Leading 3, Trailing 3, Sliding Window 4:15, MINLEN 36) (76) and quality parameters were assessed using FastQC (v0.11.7) (77). Trimmed reads were used to create genome assemblies with SPAdes (v3.12.0) (78). Assembly statistics were assessed using QUAST (v4.6.3) (79), BBMap (v38.08) (80), and SAMtools (v0.1.8) (81). Genome assemblies were checked to ensure that lengths (2.8 to 3.7 Mb) and G+C contents (34 to 45%) were consistent with those expected for *Listeria* spp. (29, 30) and of sufficient quality (number of contigs <100 and read coverage >50×). Assemblies not characteristic of *Listeria* spp. (considering genome length, G+C content, ribosomal MLST [rMLST], ANI, etc.) (*n* = 6) were excluded from further analyses (those were identified as belonging to other genera, including *Bacillus*, *Lactococcus*, *Lysinbacillus*, and *Niallia*). Assembly completeness and quality were additionally assessed using CheckM (v1.1.3; lineage-specific workflow) (31). Assemblies were annotated with the NCBI Prokaryotic Genome Annotation Pipeline (PGAP) (v5.3).

**Species identification and characterization.** For species-level identification, PYANI (v0.2.11) (82) was used to calculate ANI between isolates and type strains (downloaded from NCBI RefSeq) (see Table S2 in the supplemental material), and bactaxR (v 0.1.0) (83) was used to create an ANI dendrogram. Currently, the only *L. marthii* type assembly on NCBI is listed as suppressed from RefSeq ("excluded from RefSeq: fragmented assembly"); the GenBank version of the type strain assembly was included, as well as some additional *L. marthii* genome assemblies. *Brochothrix* spp. type strains were also included as an outgroup. Isolate pairs with a high ANI (>99.99%) were further evaluated using the CFSAN SNP Pipeline (v 1.0.1) (84). Species-level identifications were also evaluated using the rMLST tool available from PubMLST (85, 86) and the Type Strain Genome Server (TYGS) (87). Assemblies were submitted to the BIGSdb-Lm platform of Institut Pasteur to determine MLST ST and clonal complex (CC), lineage, and serogroup (36, 45, 88). The NCBI Pathogen Detection Isolates browser (89) was used to determine if the *L. monocytogenes* isolates were closely related to any existing isolates in that database. For all isolates, the Similar Genome Finder program on PATRIC (v3.6.12) (90) and BLAST (91) (using the largest contig or a 1-Mb portion of the largest contig as the query and the megablast algorithm) were used to find closely related isolates (Table S2), then PYANI was used to determine ANI between the assemblies and their most similar strains. VirulenceFinder (92) was used to identify virulence-associated genes (using minimum thresholds of 85% ID and 60% length), along with the results from the BIGSdb-Lm platform (default parameters).

**Data availability.** Raw sequencing reads were submitted to the Sequencing Read Archive and annotated genome assemblies were submitted to GenBank on NCBI under BioProject PRJNA760531. BioSample IDs are listed in Table S1 in the supplemental material.

## SUPPLEMENTAL MATERIAL

Supplemental material is available online only.
**SUPPLEMENTAL FILE 1**, XLSX file, 0.2 MB.
**SUPPLEMENTAL FILE 2**, XLSX file, 0.03 MB.

## ACKNOWLEDGMENTS

This research was supported by the University of Tennessee's Office of Undergraduate Research Summer Undergraduate Research Internship Program. We also acknowledge Geographic Information Systems (GIS) Specialist Eric Arnold with the University of Tennessee Knoxville Libraries for his help with creating the maps.

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
