## [Reviewer comments · Microbiology Spectrum]

Microbiology Spectrum

Soil collected from a single Great Smoky Mountains trail contains a diversity of *Listeria monocytogenes* and *Listeria* spp.

Michelle Claxton, Lauren Hudson, Daniel Bryan, and Thomas Denes

Corresponding Author(s): Thomas Denes, The University of Tennessee Knoxville

Review Timeline:

Submission Date:	April 19, 2022
Editorial Decision:	August 16, 2022
Revision Received:	September 30, 2022
Editorial Decision:	October 24, 2022
Revision Received:	November 3, 2022
Accepted:	November 29, 2022

Editor: Erik Hom

Reviewer(s): The reviewers have opted to remain anonymous.

Transaction Report:

DOI: <https://doi.org/10.1128/spectrum.01431-22>

August 16, 2022

Dr. Thomas G Denes
University of Tennessee
Food Science
2600 River Drive
Knoxville 37996-3591

Re: Spectrum01431-22 (Soil collected from a single Great Smoky Mountains trail contains a diversity of *Listeria monocytogenes* and *Listeria* spp.)

Dear Dr. Thomas G Denes:

Thank you for submitting your manuscript to Microbiology Spectrum. A couple of reviewers found your manuscript interesting and worth publishing provided you make some revisions. Please make sure to address the reviewers' comments in the text of your manuscript, either making changes/revisions as needed or incorporating text to address the point made. When submitting the revised version of your paper, please provide (1) point-by-point responses to the issues raised by the reviewers as file type "Response to Reviewers," not in your cover letter, and (2) a PDF file that indicates the changes from the original submission (by highlighting or underlining the changes) as file type "Marked Up Manuscript - For Review Only". Please use this link to submit your revised manuscript - we strongly recommend that you submit your paper within the next 60 days or reach out to me. Detailed instructions on submitting your revised paper are below.

Link Not Available

Sincerely,

Erik Hom

Journals Department
Reviewer comments:

Reviewer #3 (Comments for the Author):

Summary:

What is the main message of the paper?

The primary goal of this study was to isolate *Listeria* spp. from native soil. The genomes of 17 *Listeria* isolates were compared to each other and those in public databases to determine if they were novel. The site chosen for sampling was located within the

most biodiverse of all national parks in the US. The significance of the study is its potential impact on improving food safety, as well as its contribution to the understanding of *Listeria* spp evolution. In total, 17 *Listeria* spp isolates were sequenced and analyzed, 14 of which were distinct, and all of which were novel.

Are you convinced that the data presented supports the main conclusions?

Overall, the data presented supports their main conclusions. The methods used to sequence and assemble the genomes are appropriate, as are the methods for genome comparison. Their claim that the results contribute to our understanding of *Listeria* evolution would be better supported with additional metadata and more advanced genome annotations.

Major points:

1. Certain metadata were collected (site location coordinates, terrain description, time, and sample mass). The inclusion of other metadata, such as soil organic matter, moisture, pH, and other chemistry, could help to describe how these species contribute to the ecological landscape and their potential as food pathogens.
2. The authors make the statement that future studies should include sampling from a larger and broader geographical region to better understand the evolutionary characteristics of *Listeria* spp. Annotating these 17 genomes could contribute to this understanding at the current time.
3. The claim that the *Listeria* present in these soils could be transported downstream to a more heavily populated area may be a bit exaggerated, especially since their survival in surface waters, ability to persist and reproduce in water, and pathogenic capability, are unknown. Advanced genome annotation may support this claim, particularly if virulence factors and antibiotic resistance genes are analyzed.

Minor points:

1. There are some instances where the use of italics is inconsistent (ex. line 75, 198).
2. The statement made in lines 78-79 is unclear. I am not sure if the strains to which they refer are those identified in this current study or those reported in the literature.

Reviewer #4 (Comments for the Author):

Dear authors,

thank you for the submission of this rather interesting manuscript. The sources and possible natural reservoirs of microorganisms that are potential human pathogens deserve indeed some attention. The manuscript is rather short (which is not in itself a bad thing), but in some parts could be a bit more detailed. Especially for non-experts in *Listeria* spp., like myself and probably also many future readers, some more explanation and background information would be beneficial to the manuscript.

So here are some general comments on the manuscript:

The last part of the introduction should ideally lead up to the research questions or aims. However, here you first mention the intention of the study (line 82) and then continue with more background information. Restructuring the last part of the introduction by moving your research aims to the end will make the text more streamlined and consistent.

In your study, you isolated *Listeria* spp. from soil samples. However, the assumption is that these strains might reach the river and thus be transported downstream. Did you also have a look at the water in the nearby stream to see if you actually can find the same strains? That would be a necessary pre-condition if you want to assume that these soils might be reservoirs for pathogens in downstream waters.

One thing that did not become clear to me: You detected a variety of *Listeria* spp. strains, but as you state in line 164pp, none of them were closely related to those isolated from food, food processing or clinical settings, which raises the question: If they are really different, wouldn't the conclusion rather be, that soils and other environments are not the source for food-borne *Listeria* outbreaks and that their sources need to be looked for elsewhere? How likely are those *Listeria* spp. that you isolated to be pathogenic? Are they human pathogens or maybe rather animal pathogens?

Related to the previous point: How extensive was your isolation approach, did you e.g., do replicates? How likely is it, that you would have missed some important and potentially pathogenic *Listeria* with your approach?

You did whole genome sequencing of your strains, but as far as I can tell from the manuscript, you merely used the sequenced genomes for identification. Would it not be worthwhile to explore the genetic potential of your isolated strains and check for genes and/or pathway that are known to be important for pathogenic *Listeria*? This might provide more insight of the potential risk from those strains, even though they are not closely related to *Listeria* in food-borne outbreaks.

As someone not familiar with *Listeria*, I found the description of different sero- and phylotypes a bit confusing. E.g., in lines 132pp you state that you found similar phylotypes as were found in an earlier study and how those were distributed. But what

does that actually mean? Are those phylotypes known to be common environmental *Listeria*, closely related to human or animal pathogens, do they have specific traits? And continuing from that, what factors might explain the distribution of phylotypes in different areas, why are some more widely distributed than others? Especially as you have the whole genome sequences and thus information about the genetic potential of the strains, is there any indication why the strains might be distributed the way they are?

Some more technical issues:

- In *Microbiology Spectrum*, the Results and Discussion section is presented before the Material and Methods section. Thus, all abbreviations should be explained when they first appear (i.e., in the Results and Discussion). Also, some introductory sentence for each part of the Results and Discussion is needed to briefly state the experiment from which the results were obtained
- In the text and in Fig. 1, elevation and distance are given in miles/feet. Could you convert this to metric units?
- In Fig. 1, especially in panel B, the contour lines need to be adjusted. At present, the reader does not gain any information from these contours, as the contour interval is not given anywhere and none of the contour lines are marked with an elevation (e.g., contour interval 5 m, every 25 or 50 m contour line marked). In B, the contour interval should be increased (e.g., to 20 m if original was 5 m). That way the reader could get a better idea of how the terrain was shaped, how steep the slopes are etc.

Staff Comments:

Preparing Revision Guidelines

For complete guidelines on revision requirements, please see the journal Submission and Review Process requirements at <https://journals.asm.org/journal/Spectrum/submission-review-process>. **Submissions of a paper that does not conform to *Microbiology Spectrum* guidelines will delay acceptance of your manuscript.** "

Please return the manuscript within 60 days; if you cannot complete the modification within this time period, please contact me. If you do not wish to modify the manuscript and prefer to submit it to another journal, please notify me of your decision immediately so that the manuscript may be formally withdrawn from consideration by *Microbiology Spectrum*.

Thank you for submitting your paper to *Microbiology Spectrum*.

RESPONSE TO REVIEWER COMMENTS

REVIEWER #3 (COMMENTS FOR THE AUTHOR):

Summary:

What is the main message of the paper?

The primary goal of this study was to isolate *Listeria* spp. from native soil. The genomes of 17 *Listeria* isolates were compared to each other and those in public databases to determine if they were novel. The site chosen for sampling was located within the most biodiverse of all national parks in the US. The significance of the study is its potential impact on improving food safety, as well as its contribution to the understanding of *Listeria* spp evolution. In total, 17 *Listeria* spp isolates were sequenced and analyzed, 14 of which were distinct, and all of which were novel.

Are you convinced that the data presented supports the main conclusions?

Overall, the data presented supports their main conclusions. The methods used to sequence and assemble the genomes are appropriate, as are the methods for genome comparison. Their claim that the results contribute to our understanding of *Listeria* evolution would be better supported with additional metadata and more advanced genome annotations.

General response: We thank the reviewer for comments. We address some of these items under comments 3.2 and 3.3.

Major points:

3.1. Certain metadata were collected (site location coordinates, terrain description, time, and sample mass). The inclusion of other metadata, such as soil organic matter, moisture, pH, and other chemistry, could help to describe how these species contribute to the ecological landscape and their potential as food pathogens.

This is a good suggestion. In future studies, we can collect this data and use it in our analyses. We added this statement after the mention of future studies in the conclusions paragraph: "Furthermore, additional data can be gathered, such as moisture, pH, or soil organic matter, to determine their relationship with *Listeria* spp. presence" (lines 236-238).

3.2. The authors make the statement that future studies should include sampling from a larger and broader geographical region to better understand the evolutionary characteristics of *Listeria* spp. Annotating these 17 genomes could contribute to this understanding at the current time.

We annotated the genomes with the NCBI PGAP. We added that detail to the methods section (line 289) and clarified that annotated genomes are available under data availability section (line 312).

3.3. The claim that the *Listeria* present in these soils could be transported downstream to a more heavily populated area may be a bit exaggerated, especially since their survival in surface waters, ability to persist and reproduce in water, and pathogenic capability, are unknown. Advanced genome annotation may support this claim, particularly if virulence factors and antibiotic resistance genes are analyzed.

We have added some details to the discussion section describing the ability of *Listeria* spp. to be transported to water systems and their survival in water, along with references (lines 191-193). We were sure to not overstate this speculation, e.g., by the use of the words “could” and “potentially.” Additionally, we have added an analysis to identify virulence genes (lines 166-175, 312-313).

Minor points:

- 3.4. There are some instances where the use of italics is inconsistent (ex. line 75, 198).
This was corrected in both places (lines 75, 258) and the manuscript was checked for other instances.
- 3.5. The statement made in lines 78-79 is unclear. I am not sure if the strains to which they refer are those identified in this current study or those reported in the literature.
The sentence was modified and now reads: “The geographical distance between strains has also been shown to correlate with gene content difference in other genera, but only minimally in *Listeria* spp.” (lines 84-85).

REVIEWER #4 (COMMENTS FOR THE AUTHOR):

Dear authors,
thank you for the submission of this rather interesting manuscript. The sources and possible natural reservoirs of microorganisms that are potential human pathogens deserve indeed some attention. The manuscript is rather short (which is not in itself a bad thing), but in some parts could be a bit more detailed. Especially for non-experts in *Listeria* spp., like myself and probably also many future readers, some more explanation and background information would be beneficial to the manuscript.

Response: Thank you for the comments. We have added some additional information to the intro, see specific comments 4.7.

So here are some general comments on the manuscript:

- 4.1. The last part of the introduction should ideally lead up to the research questions or aims. However, here you first mention the intention of the study (line 82) and then continue with more background information. Restructuring the last part of the introduction by moving your research aims to the end will make the text more streamlined and consistent.
This portion has been edited based on your suggestion (lines 88-111).
- 4.2. In your study, you isolated *Listeria* spp. from soil samples. However, the assumption is that these strains might reach the river and thus be transported downstream. Did you also have a look at the water in the nearby stream to see if you actually can find the same strains? That would be a necessary pre-condition if you want to assume that these soils might be reservoirs for pathogens in downstream waters.

We were sure to not overstate this speculation, e.g., by the use of the words “could” and “potentially.” Additionally, we have added some details to the discussion section describing the ability of listeria spp. to be transported to water systems and its survival in water, along with references (lines 191-193); these give further support to the theory. However, this warrants further study to confirm in this specific situation, which would be challenging and a major undertaking.

- 4.3. One thing that did not become clear to me: You detected a variety of *Listeria* spp. strains, but as you state in line 164pp, none of them were closely related to those isolated from food, food processing or clinical settings, which raises the question: If they are really different, wouldn't the conclusion rather be, that soils and other environments are not the source for food-borne *Listeria* outbreaks and that their sources need to be looked for elsewhere? How likely are those *Listeria* spp. that you isolated to be pathogenic? Are they human pathogens or maybe rather animal pathogens?

We cannot draw that conclusion based on the scope of this research study, as there is the potential that other soil isolates may be sources for illnesses or outbreaks. Additionally, as we state in the manuscript, it may be valuable to have these isolates sequenced and available in public databases so that their relatedness to future illness/outbreak isolates can be assessed. We added an analysis to identify virulence genes (lines 166-175, 312-313).

- 4.4. Related to the previous point: How extensive was your isolation approach, did you e.g., do replicates? How likely is it, that you would have missed some important and potentially pathogenic *Listeria* with your approach?

We took 12 samples on the same day in the same region and the enrichment and isolation targeted both *L. monocytogenes* and *Listeria* spp. This was exploratory sampling to determine the potential diversity of *Listeria* that can be found in this region. In future studies, it would be interesting to look for associations with certain variables/environmental factors (e.g., elevation, season, etc.) and, in that situation, we would choose replicates for each condition.

- 4.5. You did whole genome sequencing of your strains, but as far as I can tell from the manuscript, you merely used the sequenced genomes for identification. Would it not be worthwhile to explore the genetic potential of your isolated strains and check for genes and/or pathway that are known to be important for pathogenic *Listeria*? This might provide more insight of the potential risk from those strains, even though they are not closely related to *Listeria* in food-borne outbreaks.

From the sequencing data, we assembled and annotated the genomes, and then determined genotypes (serogroup, MLST, cgMLST) and average nucleotide identity (ANI), which provide information on subtypes and relatedness. Also, we have submitted these annotated *Listeria* spp. genomes to a public database (NCBI), along with details on their collection location and how they were collected; we believe this is of value for the scientific community. Two of the identified strains were determined to belong to a novel species (after the initial submission of this work); this information was added (lines 146-148).

Additionally, we have added an analysis to identify virulence genes (lines 166-175, 312-313).

- 4.6. As someone not familiar with *Listeria*, I found the description of different sero- and phylotypes a bit confusing. E.g., in lines 132pp you state that you found similar phylotypes as were found in an earlier study and how those were distributed. But what does that actually mean? Are those phylotypes known to be common environmental *Listeria*, closely related to human or animal pathogens, do they have specific traits? And continuing from that, what factors might explain the distribution of phylotypes in different areas, why are some more widely distributed than others? Especially as you have the whole genome sequences and thus information about the genetic potential of the strains, is there any indication why the strains might be distributed the way they are?

Additional information about the phylogroups was added into the introduction (lines 74-83).

Some more technical issues:

- 4.7. In *Microbiology Spectrum*, the Results and Discussion section is presented before the Material and Methods section. Thus, all abbreviations should be explained when they first appear (i.e., in the Results and Discussion). Also, some introductory sentence for each part of the Results and Discussion is needed to briefly state the experiment from which the results were obtained

We have edited the manuscript so that abbreviations were spelled out the first time they were introduced in the Results and Discussion section. Introductory sentences have been added into the Results and Discussion section (e.g., lines 149, 166).

- 4.8. In the text and in Fig. 1, elevation and distance are given in miles/feet. Could you convert this to metric units?

Values in the text were converted to metric units (lines 105, 195). In Figure 1, we re-made the maps with a 25 m contour interval (see response to comment 4.10) and both maps now have a scale bar with both km and miles.

- 4.9. In Fig. 1, especially in panel B, the contour lines need to be adjusted. At present, the reader does not gain any information from these contours, as the contour interval is not given anywhere and none of the contour lines are marked with an elevation (e.g., contour interval 5 m, every 25 or 50 m contour line marked). In B, the contour interval should be increased (e.g., to 20 m if original was 5 m). That way the reader could get a better idea of how the terrain was shaped, how steep the slopes are etc.

We re-made the maps using a different contour dataset with 25 m intervals. The interval was added to the figure legend (line 331). The maps now have a key for contour elevation ranges, and elevations have been added to some of the contours on the map.

October 24, 2022

Dr. Thomas G Denes
University of Tennessee
Food Science
2600 River Drive
Knoxville 37996-3591

Re: Spectrum01431-22R1 (Soil collected from a single Great Smoky Mountains trail contains a diversity of *Listeria monocytogenes* and *Listeria* spp.)

Dear Dr. Thomas G Denes:

Thank you for submitting your manuscript to Microbiology Spectrum. Two reviewers have read over your manuscript and generally agree that it is much improved. There are a few queries and small revisions they request, with which I concur. For example, please include more details about the virulence factors you have identified.

Link Not Available

Sincerely,

Erik Hom

Journals Department
Reviewer comments:

Reviewer #3 (Comments for the Author):

Summary:

This study aimed to examine the diversity of *Listeria* spp. strains from the soil in a small region of the Great Smoky Mountains National Park. In total, 17 *Listeria* spp isolates were sequenced and analyzed, 14 of which were distinct. The data analysis supports that *Listeria* retrieved from these samples is diverse. I appreciate that the authors understand this is an exploratory

project and their results may have implications for understanding the evolution of *Listeria*. All methodologies and analyses are appropriate.

Major points:

1. Were the 17 *Listeria* isolates the only isolates that were retrieved from the enrichment? If not, how many potential *Listeria* colonies were present? A statement that other phylogroups were not isolated in this study (line 162) might have been due to sampling size if only a subset of *Listeria* were analyzed.
2. I suggest the authors include a supplementary file that lists all the virulence factors identified per isolate. Were they all chromosomally borne, or were any identified on a mobile genetic element?
3. Did your assembly strategy include the identification of plasmids or other mobile genetic elements and the presence of virulence factors on them?

Minor points:

1. Line 83-84 is incomplete ("Other researchers have found that *Listeria* spp.")
2. I appreciate the inclusion of the virulence factor analysis. Why were these chosen rather than something like antibiotic-resistance genes? Both have implications for the potential of these isolates to be pathogens. A statement of why virulence factors were chosen to analyze would be helpful. A bit more detail regarding why those *Listeria*-specific virulence factors are of importance will help readers understand the significance of examining the diversity of this genus.
3. In Materials and Methods, please clarify the *Listeria* colony morphology that was used to choose isolates for further study.

Reviewer #4 (Public repository details (Required)):

Sequencing data, has been deposited by the authors to appropriate repository.

Reviewer #4 (Comments for the Author):

Dear authors,

thank you for addressing the issues raised by reviewer 3 and myself. The manuscript has improved significantly and is now easier to understand, especially for people outside of the field of *Listeria* research. The addition of the virulence gene analysis is very valuable. I would, however, ask that you provide a table a the virulence genes (in supplement) and in which isolates they were found. Apart from that I do not have any further requests for change.

Staff Comments:

Preparing Revision Guidelines

Please return the manuscript within 60 days; if you cannot complete the modification within this time period, please contact me. If you do not wish to modify the manuscript and prefer to submit it to another journal, please notify me of your decision immediately so that the manuscript may be formally withdrawn from consideration by Microbiology Spectrum.

RESPONSE TO REVIEWER COMMENTS

REVIEWER #3 (COMMENTS FOR THE AUTHOR):

Summary:

What is the main message of the paper?

The primary goal of this study was to isolate *Listeria* spp. from native soil. The genomes of 17 *Listeria* isolates were compared to each other and those in public databases to determine if they were novel. The site chosen for sampling was located within the most biodiverse of all national parks in the US. The significance of the study is its potential impact on improving food safety, as well as its contribution to the understanding of *Listeria* spp evolution. In total, 17 *Listeria* spp isolates were sequenced and analyzed, 14 of which were distinct, and all of which were novel.

Are you convinced that the data presented supports the main conclusions?

Overall, the data presented supports their main conclusions. The methods used to sequence and assemble the genomes are appropriate, as are the methods for genome comparison. Their claim that the results contribute to our understanding of *Listeria* evolution would be better supported with additional metadata and more advanced genome annotations.

General response: We thank the reviewer for comments. We address some of these items under comments 3.2 and 3.3.

Major points:

3.1. Certain metadata were collected (site location coordinates, terrain description, time, and sample mass). The inclusion of other metadata, such as soil organic matter, moisture, pH, and other chemistry, could help to describe how these species contribute to the ecological landscape and their potential as food pathogens.

This is a good suggestion. In future studies, we can collect this data and use it in our analyses. We added this statement after the mention of future studies in the conclusions paragraph: "Furthermore, additional data can be gathered, such as moisture, pH, or soil organic matter, to determine their relationship with *Listeria* spp. presence" (lines 236-238).

3.2. The authors make the statement that future studies should include sampling from a larger and broader geographical region to better understand the evolutionary characteristics of *Listeria* spp. Annotating these 17 genomes could contribute to this understanding at the current time.

We annotated the genomes with the NCBI PGAP. We added that detail to the methods section (line 289) and clarified that annotated genomes are available under data availability section (line 312).

3.3. The claim that the *Listeria* present in these soils could be transported downstream to a more heavily populated area may be a bit exaggerated, especially since their survival in surface waters, ability to persist and reproduce in water, and pathogenic capability, are unknown. Advanced genome annotation may support this claim, particularly if virulence factors and antibiotic resistance genes are analyzed.

We have added some details to the discussion section describing the ability of *Listeria* spp. to be transported to water systems and their survival in water, along with references (lines 191-193). We were sure to not overstate this speculation, e.g., by the use of the words “could” and “potentially.” Additionally, we have added an analysis to identify virulence genes (lines 166-175, 312-313).

Minor points:

- 3.4. There are some instances where the use of italics is inconsistent (ex. line 75, 198).
This was corrected in both places (lines 75, 258) and the manuscript was checked for other instances.
- 3.5. The statement made in lines 78-79 is unclear. I am not sure if the strains to which they refer are those identified in this current study or those reported in the literature.
The sentence was modified and now reads: “The geographical distance between strains has also been shown to correlate with gene content difference in other genera, but only minimally in *Listeria* spp.” (lines 84-85).

REVIEWER #4 (COMMENTS FOR THE AUTHOR):

Dear authors,
thank you for the submission of this rather interesting manuscript. The sources and possible natural reservoirs of microorganisms that are potential human pathogens deserve indeed some attention. The manuscript is rather short (which is not in itself a bad thing), but in some parts could be a bit more detailed. Especially for non-experts in *Listeria* spp., like myself and probably also many future readers, some more explanation and background information would be beneficial to the manuscript.

Response: Thank you for the comments. We have added some additional information to the intro, see specific comments 4.7.

So here are some general comments on the manuscript:

- 4.1. The last part of the introduction should ideally lead up to the research questions or aims. However, here you first mention the intention of the study (line 82) and then continue with more background information. Restructuring the last part of the introduction by moving your research aims to the end will make the text more streamlined and consistent.
This portion has been edited based on your suggestion (lines 88-111).
- 4.2. In your study, you isolated *Listeria* spp. from soil samples. However, the assumption is that these strains might reach the river and thus be transported downstream. Did you also have a look at the water in the nearby stream to see if you actually can find the same strains? That would be a necessary pre-condition if you want to assume that these soils might be reservoirs for pathogens in downstream waters.

We were sure to not overstate this speculation, e.g., by the use of the words “could” and “potentially.” Additionally, we have added some details to the discussion section describing the ability of listeria spp. to be transported to water systems and its survival in water, along with references (lines 191-193); these give further support to the theory. However, this warrants further study to confirm in this specific situation, which would be challenging and a major undertaking.

- 4.3. One thing that did not become clear to me: You detected a variety of *Listeria* spp. strains, but as you state in line 164pp, none of them were closely related to those isolated from food, food processing or clinical settings, which raises the question: If they are really different, wouldn't the conclusion rather be, that soils and other environments are not the source for food-borne *Listeria* outbreaks and that their sources need to be looked for elsewhere? How likely are those *Listeria* spp. that you isolated to be pathogenic? Are they human pathogens or maybe rather animal pathogens?

We cannot draw that conclusion based on the scope of this research study, as there is the potential that other soil isolates may be sources for illnesses or outbreaks. Additionally, as we state in the manuscript, it may be valuable to have these isolates sequenced and available in public databases so that their relatedness to future illness/outbreak isolates can be assessed. We added an analysis to identify virulence genes (lines 166-175, 312-313).

- 4.4. Related to the previous point: How extensive was your isolation approach, did you e.g., do replicates? How likely is it, that you would have missed some important and potentially pathogenic *Listeria* with your approach?

We took 12 samples on the same day in the same region and the enrichment and isolation targeted both *L. monocytogenes* and *Listeria* spp. This was exploratory sampling to determine the potential diversity of *Listeria* that can be found in this region. In future studies, it would be interesting to look for associations with certain variables/environmental factors (e.g., elevation, season, etc.) and, in that situation, we would choose replicates for each condition.

- 4.5. You did whole genome sequencing of your strains, but as far as I can tell from the manuscript, you merely used the sequenced genomes for identification. Would it not be worthwhile to explore the genetic potential of your isolated strains and check for genes and/or pathway that are known to be important for pathogenic *Listeria*? This might provide more insight of the potential risk from those strains, even though they are not closely related to *Listeria* in food-borne outbreaks.

From the sequencing data, we assembled and annotated the genomes, and then determined genotypes (serogroup, MLST, cgMLST) and average nucleotide identity (ANI), which provide information on subtypes and relatedness. Also, we have submitted these annotated *Listeria* spp. genomes to a public database (NCBI), along with details on their collection location and how they were collected; we believe this is of value for the scientific community. Two of the identified strains were determined to belong to a novel species (after the initial submission of this work); this information was added (lines 146-148).

Additionally, we have added an analysis to identify virulence genes (lines 166-175, 312-313).

- 4.6. As someone not familiar with *Listeria*, I found the description of different sero- and phylotypes a bit confusing. E.g., in lines 132pp you state that you found similar phylotypes as were found in an earlier study and how those were distributed. But what does that actually mean? Are those phylotypes known to be common environmental *Listeria*, closely related to human or animal pathogens, do they have specific traits? And continuing from that, what factors might explain the distribution of phylotypes in different areas, why are some more widely distributed than others? Especially as you have the whole genome sequences and thus information about the genetic potential of the strains, is there any indication why the strains might be distributed the way they are?

Additional information about the phylogroups was added into the introduction (lines 74-83).

Some more technical issues:

- 4.7. In *Microbiology Spectrum*, the Results and Discussion section is presented before the Material and Methods section. Thus, all abbreviations should be explained when they first appear (i.e., in the Results and Discussion). Also, some introductory sentence for each part of the Results and Discussion is needed to briefly state the experiment from which the results were obtained

We have edited the manuscript so that abbreviations were spelled out the first time they were introduced in the Results and Discussion section. Introductory sentences have been added into the Results and Discussion section (e.g., lines 149, 166).

- 4.8. In the text and in Fig. 1, elevation and distance are given in miles/feet. Could you convert this to metric units?

Values in the text were converted to metric units (lines 105, 195). In Figure 1, we re-made the maps with a 25 m contour interval (see response to comment 4.10) and both maps now have a scale bar with both km and miles.

- 4.9. In Fig. 1, especially in panel B, the contour lines need to be adjusted. At present, the reader does not gain any information from these contours, as the contour interval is not given anywhere and none of the contour lines are marked with an elevation (e.g., contour interval 5 m, every 25 or 50 m contour line marked). In B, the contour interval should be increased (e.g., to 20 m if original was 5 m). That way the reader could get a better idea of how the terrain was shaped, how steep the slopes are etc.

We re-made the maps using a different contour dataset with 25 m intervals. The interval was added to the figure legend (line 331). The maps now have a key for contour elevation ranges, and elevations have been added to some of the contours on the map.

Reviewer comments:

Reviewer #3 (Comments for the Author):

Summary: This study aimed to examine the diversity of *Listeria* spp. strains from the soil in a small region of the Great Smoky Mountains National Park. In total, 17 *Listeria* spp isolates were sequenced and analyzed, 14 of which were distinct. The data analysis supports that *Listeria* retrieved from these samples is diverse. I appreciate that the authors understand this is an exploratory project and their results may have implications for understanding the evolution of *Listeria*. All methodologies and analyses are appropriate.

We thank Reviewer #3 for their constructive comments.

Major points:

3.1. Were the 17 *Listeria* isolates the only isolates that were retrieved from the enrichment? If not, how many potential *Listeria* colonies were present? A statement that other phylogroups were not isolated in this study (line 162) might have been due to sampling size if only a subset of *Listeria* were analyzed.

We have edited the text to more clearly describe how colonies were isolated (lines 239-248). As these were enrichments, each plate had multiple colonies, but only one was selected (up to two total for each soil sample, if colonies characteristic of *Listeria* spp. were present). We originally isolated and sequenced 23 isolates, but 6 of those were identified as belonging to other genera – this information was added (lines 247, 252, 267-270). We also revised that portion of the Results & Discussion section to read: “however, none of these phylogroups were isolated in the present study, which may indicate that they were not present in the samples. Alternatively, these phylogroups may have been present in the samples but not recovered; for example, they could have been suppressed or outcompeted by other strains during the enrichment step (47-53) or their colonies may have simply not been selected off of the MOX plates for isolation and characterization.” (lines 169-174).

3.2. I suggest the authors include a supplementary file that lists all the virulence factors identified per isolate. Were they all chromosomally borne, or were any identified on a mobile genetic element?

We added a supplemental file with the identified virulence factors (**File S1**) and a reference to this file in the main text (line 183).

3.3. Did your assembly strategy include the identification of plasmids or other mobile genetic elements and the presence of virulence factors on them?

For this study, we didn't perform long-read sequencing, so we do not have complete closed genomes, which makes the identification of plasmids more challenging. However, we did perform and publish a follow-up study on the novel species isolates (referenced in lines 118-120) and characterized a plasmid in one of those isolates. We would prefer to not add additional analyses at this stage in revisions, but if the editor feels that it is necessary, we could perform some analyses to identify MGE in these draft genomes.

Minor points:

3.4. Line 83-84 is incomplete ("Other researchers have found that *Listeria* spp.)

This was removed (line 83).

3.5. I appreciate the inclusion of the virulence factor analysis. Why were these chosen rather than something like antibiotic-resistance genes? Both have implications for the potential of these isolates to be pathogens. A statement of why virulence factors were chosen to analyze would be helpful. A bit more detail regarding why those *Listeria*-specific virulence factors are of importance will help readers understand the significance of examining the diversity of this genus.

The decision to focus on virulence factors was based on the initial reviewer comments.

More details about the roles of the virulence factors were added into the text (lines 142-155) and in the supplemental file (File S1).

3.6. In Materials and Methods, please clarify the *Listeria* colony morphology that was used to choose isolates for further study.

This detail was added to the manuscript: "circular black colony surrounded by a black zone" (line 246).

Reviewer #4 (Public repository details (Required)):

Sequencing data, has been deposited by the authors to appropriate repository.

Reviewer #4 (Comments for the Author):

4.1. Dear authors,

thank you for addressing the issues raised by reviewer 3 and myself. The manuscript has improved significantly and is now easier to understand, especially for people outside of the field of *Listeria* research. The addition of the virulence gene analysis is very valuable. I would, however, ask that you provide a table of the virulence genes (in supplement) and in which isolates they were found. Apart from that I do not have any further requests for change.

We thank reviewer # 4 for their comments.

We added a supplemental file with the identified virulence factors (File S1) and a reference to this file in the main text (line 183).

November 29, 2022

Dr. Thomas G Denes
The University of Tennessee Knoxville
Food Science
2600 River Drive
Knoxville 37996-3591

Re: Spectrum01431-22R2 (Soil collected from a single Great Smoky Mountains trail contains a diversity of *Listeria monocytogenes* and *Listeria* spp.)

Dear Dr. Thomas G Denes:

Thanks for your patience! I have reviewed your revisions and am glad to accept your manuscript for publication. I am forwarding it to the ASM Journals Department for publication. You will be notified when your proofs are ready to be viewed. Please make sure that your Supplementary Tables are submitted as separate files (if possible) instead of bundled into one file, especially since you have an additional Supplementary File 1 which could easily confuse a reader.

Sincerely,

Erik Hom
Editor, Microbiology Spectrum

Journals Department
Supplemental Tables: Accept
Supplemental File 1: Accept